# Healthcare Professionals’ Experiences with Patient Participation in a Mental Healthcare Centre: A Qualitative Study

**DOI:** 10.3390/ijerph20031965

**Published:** 2023-01-20

**Authors:** Kim Jørgensen, Mathias Hansen, Trine Groth Andersen, Morten Hansen, Bengt Karlsson

**Affiliations:** 1Department of Public Health, Nursing, Aarhus University, 8000 Aarhus C, Denmark; 2University College Diaconissestiftelsen, 2000 Frederiksberg, Denmark; 3Psychiatric Outpatient Clinic Ishøj, Bostedsteamet, 2635 Ishøj, Denmark; 4Centre for Mental Health and Substance Abuse, Department of Health, Social and Welfare Studies, Faculty of Health and Social Sciences, University of Southeastern Norway, P.O. Box 7053, 3007 Drammen, Norway

**Keywords:** patient participation, mental health, psychosocial nursing, holistic care, communication, compliance

## Abstract

Introduction: Patient participation is a cornerstone of the debate concerning healthcare professionals and patients of mental health centres. It constitutes an objective in government health policy in Scandinavia and other Western countries. However, little is known about the experiences of healthcare professionals in mental healthcare practices involving patients under their treatment and care. Objective: This study aimed to explore the experiences of healthcare professionals with patient participation in the context of a mental health centre. Methodological design: Four focus group interviews with healthcare professionals reflected differing experiences with unfolding patient participation in clinical practices in four wards of a mental health centre. A content analysis developed and framed themes. Results: Patient participation was based on structural conditions, which shows that predetermined structural methods predominantly control involvement. The structural methods are seen as promoting participation from the patient’s perspective. At the same time, the methods also enable taking account of the individual patient’s wishes and needs for involvement. Discussion and conclusion: This study illuminates the meaning of patient participation in a mental health centre based on the social interactions among nurses and other healthcare professionals. The approach can contribute to dealing with the challenges of incorporating patient participation as an ideology for all patients in a psychiatric context, which is important knowledge for healthcare professionals.

## 1. Introduction

In many Western countries, in recent decades, patient participation has been a milestone in the health sector. The introduction of patient participation has given patients a new identity as active and decision-making participants in care [1,2]. Existing research shows how patient participation can be understood from different perspectives. The meaning of patient participation in mental healthcare can be understood as different discourses. In the collection of discourses about patient participation, we see a policy discourse, a consumer discourse, and a biomedical discourse [3,4]. The politics and consumer discourses frame healthcare as an individual approach to care, and the biomedical discourse constrains a paternalistic focus on diagnoses and symptoms. These positions affect how healthcare professionals will meet the patient and how they must participate in healthcare in the setting of a mental health centre [5,6].

Involvement in one’s care is a civil right. In Danish law, the patient’s right to self-determination must be respected. Therefore the patient must have information about their illness and treatment so as to make choices and give consent before treatment is initiated [7]. The issue of patient participation is, therefore, not up for discussion but a goal that both healthcare professionals and patients must face and implement [7,8,9,10,11]. In a study, patient participation is equated with, among other things, the dissemination of knowledge to the patient [5]. In other studies, neoliberal discourse frames patient participation as a vague concept. On one hand, patient participation reflects a democratic right to influence one’s own treatment, which can be perceived as positive. On the other hand, patient participation can also be seen as an individualization of responsibility for the patient’s own health [1]. The more the patient takes responsibility for their own situation and health, the less they use health services [1,12,13]. Politically, there can, therefore, also be an economic incentive to make patient participation an ideal in the healthcare system. The healthcare system can be perceived as a company that is governed by a new public management way of thinking. The healthcare system must offer services to the customers (patients) and they must be involved and engage in their treatment so that they achieve as much self-care as possible and become independent of professional help [2,14,15].

The leading healthcare policies set up some barriers to creating a free and individually focused healthcare system [16,17]. The limitations lie in the requirements that the allocation of treatment requires a medical diagnosis. As far as possible, the treatment must be evidence-based, effective, standardized, and offered in package courses based on reference programs, i.e., general requirements, which contrast with building treatment on situational bases [18,19]. As a counterpart to this form of standardization, there is a concrete situational meeting between patient and health professional. The situational basis opens up an understanding of what happens between people as a phenomenon that unfolds in unique situations in a mental health centre. Attention must be paid to understanding encounters and human reactions in unique contexts and surroundings. The research lacks clarity about the concept of ‘patient participation’ [5,20,21,22,23]. The lack of attachment to a theoretical framework has led to poor understanding and communication among researchers, healthcare professionals, and politicians, along with problems with measurement and comparison among studies across different hospitals [24,25]. This study explores the experiences of healthcare professionals with patient participation in a mental health centre context.

## 2. Methods

### 2.1. Sampling

We used purposive sampling [26] to ensure the recruitment of participants who could provide in-depth and detailed information about the phenomenon under investigation. The first author sent information about the study and elaborated on it at meetings with management and potential participants in the study. The first author informed the participants about the purpose of the project and their legal and ethical rights. The first author came to an agreement with the manager of the mental health centre in Copenhagen. Four wards from the mental health centre, with several healthcare professionals, gave informed consent to participate in the study. Three inpatient wards and one outpatient department were included. Nurses, assistant nurses, physiotherapists, occupational therapists, social workers, and a teacher who had first-hand contact with the patient participated. Twenty-four participants gave their informed consent. The average age was approximately 40 years, and the majority were nurses with five to 20 years of mental healthcare experience; the majority had specialized in further education in psychiatric nursing (Table 1). The research group consisted of five professionals. Two researchers were trained psychiatric nurses with PhDs, a teacher with personal experience in psychiatry, and two registered nurses (RNs) took on the role of research assistants.

### 2.2. Focus Group

We conducted four focus group interviews to create a dynamic and inspiring dialogue on the experiences of healthcare professionals with patient participation in a mental health centre context. The focus groups contributed to the nuanced and rich exchanges of experiences in a way that improved data quality and produced deeper and richer data more quickly than individual interviews could have [27,28]. In the focus groups, the aim was to facilitate discussion with four to 10 participants at a time about their experiences working with patient participation [29]. The discussions were audio-recorded and transcribed verbatim. The focus group interviews were carried out in the mental health centre. Inspired by existing knowledge in the field [30,31], we formulated a list of questions to explore the experiences of our healthcare professionals in mental healthcare. Based on the list, we asked a few broad questions [32] and focused on follow-up questions to achieve richness and depth (Table 2).

### 2.3. Data Analysis

We aimed for and achieved data saturation by conducting a systematic qualitative content analysis. We developed themes to ensure this study’s validity [33]. We used manifest content analysis to conduct a low-level analysis of the experiences of healthcare professionals in mental healthcare. The systematic process consisted of four steps: condensed meanings, categories, subthemes, and themes. We read the transcribed interviews across the sample and identified meaning units, which we subsequently coded.

The code sheet was developed in an iterative process. The transcribed interviews were discussed in the research group to develop the code and to categorize main themes and subthemes. The developed themes formed the basis for a coherent understanding and were viewed in relation to the aim of the study [33]. We used NVivo software to organize the analysis [34]. The themes reflected a new understanding of concrete practices.

### 2.4. Ethical Considerations

Healthcare professionals gave their informed consent to participate in the study after receiving oral and written information about its purpose. The study was accepted by the Danish Capital Region Data Protection Agency (j.nr.: P-2020-345). It was carried out according to the Helsinki Declaration [35] and Danish law [36]; no formal permit from a biomedical ethics committee was required as the research purpose was not to influence the healthcare professionals, physically or psychologically.

## 3. Results

The focus group interviews were facilitated through the interview questions, using the social constructivism approach. All healthcare professionals were active, contributory, and listened to each other, and the understandings and opinions developed during the interviews. The main theme, ‘patient participation based on structural conditions’, was based on two themes and four subthemes. The first theme was ’patient participation is not a free choice’, with subthemes ‘patients are included’ and ‘patient participation is carried out based on structural methods and professional expectations’. The next theme was ‘patient participation is a very unclear concept’ with subthemes ‘patient participation is an individual thing’ and ‘nonparticipation’ (Table 3).

### 3.1. Patient Participation Is Not a Free Choice

In the focus group interviews, it became clear that all of the healthcare professionals had preconceived notions of what patient participation meant. It implies differing degrees of participation, ranging from giving information to patients to having them participate in meetings dealing with patient treatment. Conversely, participation is not an individual choice. Patient participation is not a free choice because it must meet the structural framework of the organization and the needs of the individual patient. Structurally predetermined treatment methods and expectations for patients to be active and compliant show that participation is governed by certain framework conditions in psychiatry that impacts the extent to which a patient is involved.

“After all, we have developed patient plans, and we work with (cognitive behavioural therapy), where everything starts with a problem–aim list, meaning that the patients must indicate what problems they have and what the goal is. We have become better at speaking from the patient’s perspective in our morning meetings, but it is enormously difficult to implement participation. It is bound by things we must do, ways we must do things, e.g., criteria for when a patient is well enough to be discharged, when they are stable enough”.(Psychomotor therapist.)

### 3.2. Patients Are Included

The healthcare professionals agreed that patients are included in the entire treatment process. In the discussion, there were examples of how they included the patients’ wishes for activities based on their personal goals.

“They are very involved, of course. So, we have this offer, and they have some goals, and then it’s about matching their goals with what we have in terms of activities and groups. So, it’s not like everyone must go through the same machine, but that they choose the groups and the activities, and if someone has the desire to get in better physical shape, then there will be many physical groups on the schedule. If someone works with social skills, there are meetings and self-esteem groups, and other things that they take part in. So, we like to talk together with patients about the weekly schedule that they each make, the treatment plan, and what we work with. Currently, they match their goals with our offer”.(Nurse.)

The professionals had a self-understanding that they meet all patients’ needs for involvement, regardless of who they are and what they may want. In the discussion, it emerged that the health professionals’ expectations for participation varied. We asked: When are patients involved?

“Well, I haven’t sat and looked at definitions of one and the other, but you can say that if you participate in a treatment plan meeting and you are there, then you have, in a sense, participated. But if you do not agree in any way with the points in the treatment plan, have you been involved? Yes, you are at least informed, but can you recognize yourself in it? No, you might not be able to”.(Nurse.)

Different preconceptions were revealed through the healthcare professionals’ discussions, which placed patient participation in a continuum between the patient being well-informed about his treatment to being involved in all decisions relating to hospitalization and the future.

### 3.3. Patient Participation Is Carried out Based on Structural Methods and Professional Expectations

Regardless of the extent to which the healthcare professionals thought they involved patients, they kept coming back to the fact that there are also structural methods and professional expectations that influence how and how much they involve patients.

“There are some criteria for you to be hospitalized. There are some things we must do here. You are manic, so it may well be that you have an idea that you would like to go out dancing for a whole weekend. Here, professionalism just comes into play again. You can’t do that. Here it is that we limit you. So, it’s as if their goals are not being met now because we know that if we let the patient go out dancing all weekend, he will be even more manic when he comes back, or even worse”.(Nurse.)

“Mental health disorders can be experienced as a challenge when healthcare professionals want to involve patients. Even if there is a will to create involvement, the framework can make it difficult to meet patients’ needs. Economy, efficiency requirements, focus on utilization of beds, and requirements for rapid investigation and treatment are examples of management tools that can challenge the individual involvement of patients. We must be realistic. They may have some hopes or dreams, so it is also our task to be quite realistic, and it may well be that we do not completely agree sometimes with the framework within which we must work. But you must accept that this, this is the framework”.(Nurse.)

At admission, an assessment of the patient takes place to decide whether hospitalization can contribute to an improvement in their condition. Here, it is also assessed whether the patient can participate in the treatment.

“Patients are assessed as suitable, and we look at what we can offer in terms of treatment and activities and whether it is something patients can participate in, and then some goals must be set”.(Psychomotor therapist.)

Many methods were discussed, and there was great confidence that evidence, medicine, cognitive behavioural therapies, and environmental therapies are the best methods that ensure consideration of the patient’s perspective.

### 3.4. Patient Participation Is a Very Unclear Concept

The healthcare professionals predominantly based their understanding of patient participation on their experiences in practice. Only a few healthcare professionals referred to definitions they had read. In practice, they work differently with participation; some use different synonyms for participation, e.g., ‘nursing’, ‘collaboration’, ‘inclusion’, and ‘information’.

“We don’t use the word patient participation. We call it collaboration”.(Psychomotor therapist.)

“After all, it is also in our structure that we must involve patients. Patient participation means that you must create a patient and treatment plan together with the patient”.(Nurse.)

“We are very involved in the way that we hold all treatment plan meetings with the patients”.(Nurse.)

Several healthcare professionals understood patient participation as fulfilling patients’ needs. There was a perception that when the patient physically participates in professional efforts, it means that they have participated. Situations in which the health professional and patient collaborate to solve a task together are considered participation.

### 3.5. Patient Participation Is an Individual Thing

The healthcare professionals referred to participation as an individual phenomenon, and structural methods and objectives limit individual consideration. Structural methods involve, for example, the healthcare professionals’ choice of treatment such as psychoeducation, cognitive behavioural therapy, or environmental therapy, and these methods form a framework for the patient’s treatment options.

“We have patients who would like to be involved in decisions about, for example, being hospitalized for three months because they think that this is what is needed for them to be calm enough to come home, but this does not agree with the region’s rules that we must discharge 4.3 patients per week”.(Nurse.)

Patient participation has many nuances, and there can be a difference in how healthcare professionals perceive the term. Participation can be something very concrete for patients.

“Patient participation is something concrete for the patient. For example, we go to great lengths for the pack of cigarettes they are missing or try to find staff to accompany them to their home and pick up the shirt they miss a lot, but it may be difficult to involve them in the treatment”.(Nurse.)

Patient participation requires patients to have the resources to be an active part of their treatment and familiarize themselves with their illness and treatment. Participation is expressed as familiarizing yourself with knowledge and using it to recover. In the discussion, it therefore also becomes clear that participation imposes a greater responsibility on the patients for the process.

“Patients do not always have the resources to be involved, at least not on our terms. So, we have to do it in a different way, on the patient’s terms, so that they have the resources to be able to participate”.(Nurse.)

“In most cases, patients are in voluntary treatment, and participation requires an effort on their part. There is no one sitting and doing a lot of things for you, and then you are cured, and then you can get out again. It is a culture that, fortunately, I think is leaving us in psychiatry. It actually requires a lot from them to be involved from the start, and we are clear about what we expect from them”.(Nurse.)

These views show that the degree of participation is also controlled by how many resources the patient has to participate. If fewer resources are used, there is a risk of getting less benefit from the healthcare efforts offered.

### 3.6. Nonparticipation

Not all patients want an active and participating role in their treatment process. The patient may not be aware of their illness or their need for treatment. Here, participation is a challenge. Health professionals must find other ways to motivate participation, such as in activities.

“For the most part, patients will neither be hospitalized nor treated, and participation is a challenge. Then we find other areas to include, for example, with activities”.(Nurse.)

Several healthcare professionals have found that the treatment requires participation and is full of dilemmas, as many patients cannot relate to many questions, weekly schedules, formulate their own needs, etc. Thus, the healthcare professionals also expressed scepticism about the requirement for participation that does not match the wishes of all patients. Some patients want to be allowed to be sick without having to do everything possible, not to mention having to already talk about discharge upon admission.

“We ask the patient what their main goal with the treatment is, but sometimes they actually want help for something else, e.g., losing weight, and do not want to participate in the standard treatment we offer”.(Nurse.)

Nonparticipation is defined as patients not wanting to actively participate in their treatment and wanting to leave the responsibility to the healthcare professionals.

“After all, we see a lot of patients who do not participate in the treatment because they do not show up. Hospitalization is, after all, a training ground to enable them to self-care”.(Nurse.)

In the discussion, patients were expected to be compliant because the treatment would not work without their active participation.

## 4. Discussion

This study illuminates the meaning of the experiences of healthcare professionals with patient participation in a mental health centre context. This understanding can contribute to the challenges of incorporating patient participation as an ideology in all psychiatric contexts and is, therefore, important knowledge for healthcare professionals such as nurses, assistant nurses, physiotherapists, occupational therapists, social workers, and physicians. Healthcare professionals articulated patient participation as an ambiguous concept that is difficult to translate into concrete actions in practice. They do not work with a specific definition. However, they all have in common that patient participation is carried out on the premises of the professionals, as it is about the patient’s participation in treatment. The lack of clarity on what patient participation means and how it must be implemented in clinical practice has implications for how it is exercised in the relationship between health professionals and patients [6,37]. Moreover, the consequence of the lack of precision in patient participation means that nurses and other professionals often act in ways that relate to how they each individually want to involve patients [1,5,6].

There is a self-understanding among most healthcare professionals that standardized treatment methods such as cognitive behavioural therapy, environmental therapy, psychoeducation, and medicine are examples of approaches that patients are subject to and must follow if they wish to be hospitalized and treated. At the same time, many of the healthcare professionals also criticized the treatment methods, as they are an expression of a standard treatment that does not match all patients’ individual needs for help and support. This result reflects other recent research, which also points out how patient participation has become a steering tool that must contribute to better patient satisfaction, better state of health, fewer hospital admissions, and lower costs per citizen [38,39]. As our study shows, some patients do not wish to be actively involved in the treatment and need a more tailored treatment that considers individual hopes, goals, and resources. The health professionals acknowledged that it can be difficult to meet all individual needs as they are guided by the organization’s goals, which must also satisfy financial and efficiency requirements. Previous research also shows that not all patients want to be actively involved in their treatment but are often met negatively [40]. As our study shows, patient participation is not a free choice. The professionals will always dictate what it is possible to participate in. Patients cannot decide their own treatment and care. Other research results also show how patient participation occurs based on the options and conditions that the professionals decide to offer. It may concern the available resources of the healthcare professionals and their belief in, for example, medical evidence as the most important approach to help [13].

In the study, there was disagreement about whether patient participation is for all patients. Some require more support to be involved, while others have the resources to make demands for participation. This phenomenon is also seen in other research, which speaks to a problem of inequality, where the most resourceful patients are involved more than those who are most severely affected by the disease [17,41,42,43]. Patient participation also implies a transfer of responsibility and that the healthcare professionals are willing to let go of some of the control. Patients are expected to take joint responsibility for their own treatment. This responsibility includes, for example, following the treatment and care that has been decided, and contributing to the development of the patient’s weekly plans, which describe the patient’s daily activities. Other research shows that when patient participation becomes a management tool for compliance, i.e., patients must follow predetermined rules for admission, the patient’s opportunities for individual involvement are reduced [6,44]. Despite the healthcare professionals’ open-mindedness to meet patient participation as an individual tool, they did not want to compromise the standardized methods. Patient participation unfolds in a field of tension between predetermined treatment methods that control the possibilities of involvement and the desire to meet the individual patient’s wishes, hopes, and dreams as a starting point for organizing the help. Research problematizes the tension between a structurally predetermined and an individually-tailored approach, as this form of management matches the neoliberal management mechanisms of Western societies, where the hospital aims to provide services and treatment efforts, e.g., knowledge about diseases and treatment, where it is up to the patient to accept the help positively and translate it into recovery as quickly as possible. Society expects the patient to be productive and an active citizen who takes responsibility for their own life [45,46,47].

There is a need for a paradigm in mental healthcare based, to a greater extent, on a treatment that helps to solve challenges that the patient considers to be the biggest in order to live a good and satisfying life and overcome the challenges that a mental disorder entails [1,13,15]. This study shows that not all patients want to be involved in the treatment and care. Nonparticipation covers, e.g., a group of patients who do not recognize their need for hospitalization and, for some, are hospitalized under coercion. If patients cannot see the purpose of the admission, they may not want to participate actively in the treatment and care. Nonparticipation has been sparsely explored, but studies show that some patients do not want active involvement, which may be because they are severely affected by illness and do not have the energy to make decisions based on difficult questions. In addition, there is a group that wants to leave the responsibility for the treatment to the professionals. With participation comes great responsibility and obligation that can be difficult to live up to at the same time as being affected by a mental disorder [6,12]. Health professionals have a special ethical responsibility to involve the most vulnerable patients and, despite the respect for meeting a wish for nonparticipation, this must not lead to a form of the sin of omission [48,49]. The desire for, or opposition to, participation may vary, but regardless of this, patients must not be deprived of the opportunity to influence their treatment course.

## 5. Implications

Patient participation has become an important aim in mental healthcare, especially in the past decade, but we still do not know much about what it means in a clinical practice where it must be carried out. This study contributes to different professional perspectives on what this means and how it unfolds in clinical practices. This knowledge can contribute to healthcare professionals being able to work consciously with the obstacles that may exist when wanting to introduce individual patient involvement.

## 6. Limitation

Some limitations need to be addressed. The investigation of how patient participation unfolded in a mental health centre using focus group interviews is an expression of the self-understanding of the healthcare professionals. An observational study could have given a more direct picture of how patient participation unfolded. Similarly, a patient perspective could have added another exciting angle to the results. It is also important to point out that the healthcare professionals were mainly nurses; however, other health professionals participated in the focus group interviews. Unfortunately, it was mainly the nurses and a psychomotor therapist that were most active in the discussions. We tried to encourage everyone to be active by addressing them more directly, but we were unable to get everyone to be equally active and to contribute to the discussion of the selected themes. We recognize that this situation could mean that we did not obtain all of the nuances included in the discussion.

## 7. Conclusions

Patient participation has different meanings among healthcare professionals. On one hand, there is an expectation that the patient is active throughout their treatment course and decisions. On the other hand, patient participation can be reduced synonymously because the patient is well-informed. There is an ambivalent approach to structural, standardized, evidence-based treatment options such as cognitive behavioural therapy, environmental therapy, and psychoeducation. The entire medical treatment is an offer that involves the patient’s perspectives. It is also recognized that structurally predetermined treatment measures also limit the possibility of taking into consideration all the individual patient’s needs for participation.

## Figures and Tables

**Table 1 ijerph-20-01965-t001:** The context of the participants.

Focus Groups	Context	Number of Participants	Title
1.	An acute mental health inpatient unit	5	A bachelor of psychomotor therapy and nurses
2.	An acute mental health inpatient unit	6	A physiotherapist, a social and healthcare assistant, and nurses
3.	An acute mental health inpatient unit	6	An occupational therapist and nurses
4.	Mental health outpatient unit	7	A social and healthcare assistant and nurses
	The number of participants:	24	

**Table 2 ijerph-20-01965-t002:** Interview guide.

Theme	Research Questions	Interview Questions
Patient participation	How is the concept of patient participation perceived by health professionals?	What does patient participation mean to you as a health professional? (There are several perceptions of participation, so self-perception is the basis for understanding the answers in the next theme.) Can participation be achieved? Do you have any criticism of the participation phenomenon? Is patient participation limited?
Patient participation in practice	How does healthcare ensure participation in practice for the patient? How do the healthcare staff work in a participatory way?	What does it mean to you to work in a participatory way? In what ways do you feel that you work in a participatory practice? Does the participated approach have limitations? What works/does not work?
Structure	How does healthcare structure participation in general? How is structure added to a patient process?	Can you describe a patient process and how this is structured? How is participation planned into the structure of the treatment? How is everyday life structured in the department for you and your patients?
Meaning/hopes/goals	How are meaning, goals, and hopes perceived, and what significance does this meaning have for the treatment?	What does hope mean to you, and what significance does it have for the patient process? What significance does it have for the patient to find meaning in the process and their situation? How do you help the patient set goals? How do you ensure that the patient is motivated to achieve these goals?
Examples of participation practice		Can you come up with concrete examples of how you work in a participatory way?

**Table 3 ijerph-20-01965-t003:** Subthemes, themes, and main theme.

Subthemes	Themes	Main Theme
Users are included.	Patient participation is not a free choice.	Patient participation is based on structural conditions.
Patient participation is carried out based on structural methods and professional expectations.		
Patient participation is an individual thing.	Patient participation is a very unclear concept.	
Nonparticipation.		

## Data Availability

Data sharing not applicable. No new data were created or analyzed in this study. Data sharing is not applicable to this article.

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
