# Peer review of "Healthcare Professionals’ Experiences with Patient Participation in a Mental Healthcare Centre: A Qualitative Study"

_ijerph, 2023, doi:10.3390/ijerph20031965_

Round 1

Reviewer 1 Report

In the current manuscript, the authors explore healthcare professionals’ experiences with perspectives on patient participation in the context of a mental health centre.

The study enquires a topical subject, as patient participation from service users is a milestone in the health sector.

The manuscript is consistent within itself and has a pleasant logic flow. Nonetheless, there are some issues that I wish the authors would fix to further improve the value of the article.

I suggest including a table or a figure with all the themes and sub-themes.

In my opinion themes and sub-themes are presented in a confusing way in the Results. Moreover, I suggest discussing which professionals expressed which themes/sub-themes (not just indicating in parentheses who said something). It could be interesting to discuss why a certain type of professional has a specific opinion on patient participation.

I suggest enriching the Limitations section focusing on specific limitations of the study and not only on general limitations of qualitative research based on focus groups as compared to observational studies.

Author Response

Dear reviewer

Thank you very much for your feedback to our paper. We have tried to accommodate comments and hope it meets your expectations. We arrange for prof reading before we resubmit the paper.

I suggest including a table or a figure with all the themes and sub-themes.

We completely agree and have made an overview that shows the themes.

In my opinion themes and sub-themes are presented in a confusing way in the Results.

In addition to the new table, we have worked with the headings so that it becomes clearer which themes are presented

Moreover, I suggest discussing which professionals expressed which themes/sub-themes (not just indicating in parentheses who said something). It could be interesting to discuss why a certain type of professional has a specific opinion on patient participation.

We discussed in the research group how we could best meet the criticism and found that we added a self-criticism in the limitation section

I suggest enriching the Limitations section focusing on specific limitations of the study and not only on general limitations of qualitative research based on focus groups as compared to observational studies.

We have added a more specific self-criticism to the limitation section

Reviewer 2 Report

The authors conduct a qualitative content analysis into the healthcare professionals' perspectives on patient participation in the patient’s treatment and care. This study offers an interesting insight into the complexity of requiring patients to be involved in their care. I feel as though there is a missed opportunity to comment or contribute to the discussion about healthcare professionals placing the burden on patients who have not been enabled (or have the skills) to make decisions about healthcare. While there the study touches on this idea, there is an opportunity to contribute more explicitly.

Overall, the text appears to be underdeveloped. There is a lack of explicit clarity and the relevance of some ideas is not present. The introduction attempts to establish the value of this study, but I am left wondering what the relevance is of some ideas. The results do well present the content of the focus groups and the commentary on some themes needs to be expanded. The discussion does well to pull the ideas and themes together.

This study needs revisions before publication.

Edits:

Please proofread and ensure all spelling and typos are corrected.

Line 12 ‘centre’ should be ‘centres’.

The objective sentence in the abstract does not make sense, the ‘experiences with perspectives on participation’. Please amend this sentence.

Introduction – Are there specific Western countries that are adopting patient participation in the way that is conceived of in this study? As a Southern Hemisphere healthcare system user (of a Western country), patient participation is not conceptualised in the same way as described in the study.

Line 35 – ‘any’ does not make sense in the sentence.

Line 35-36 – Please clarify the sentences. What do the authors mean by different perspectives and what is the relevance of views that are linked to different discourses? What is the relevance of discourse in relation to understanding patient participation or the study? Could the authors consider the value of the sentences and how this contributes meaning to the paragraph and the wider study?  

Line 41 ‘users’. Please use terms consistently, are the people being referred to ‘users’ or ‘patients’. Consistency in terms will enable the reader to follow the writing rather than getting caught up on who is the subject.

Line 49 – why is it a vague concept? Please consider clarifying the intended communication in this sentence.

Line 55 – too many spaces. Why would readers, patients, or healthcare professionals care that the participation discourse includes the burden of increased diseases on the state (i.e., public finances and maintaining the welfare state)? Could the authors link the sentences on lines 55 and 56 to the topic of participant discourse? Sentence on line 57 to earlier paragraph on line 56.

Line 58 too many spaces.

Line 60 ‘most possible treatment’ does not make sense. Please correct or explain.

Line 64 – which healthcare policies are the authors referring to? What might these policies have to do with patient participation?

Line 67 – what is the meaning ‘and so on’ in this sentence?

Line 69-67 – the sentence does make sense. What situation basis are the authors referring and which unique situations do they mean? Could they clarify this sentence to communicate the intended meaning more clearly? Please also clarify the subsequent sentence.

Line 72 – What research are the authors referring to when they say ‘the research lacks clarity’?

Line 77 – as per an earlier comment, the objective statement of ‘experiences with perspectives on patient participation’ does not make sense. It is unusual to see the combination of ‘experiences with perspectives’ as these are two distinct concepts and perhaps selecting either experiences or perspectives might start to communicate the objective more clearly.

Line 85 – ‘teacher’ might be a more accessible word than ‘pedagogues’ for western English speakers.

Line 86 – do the authors mean ‘users’ rather than ‘the user’ and is this ‘the users of the mental health centre’?

Line 87 – what do the authors mean ‘with each section’?

Lines 83 and 87 are repeating similar information

Please be consistent and clear about who the actor is in the sampling section. The authors shift from ‘the first author’ to ‘we’ to ‘the first author’ – this is confusing.

Line 90 start the sentence with ‘Twenty-four participants….’

Line 94 too many spaces

Line 98 – the title is misspelt. Focus group should not be plural. Please check the spaces and alignment of the table contents.  

Line 108 – why is there an apostrophe?

In lines 108 and 109 there is a shift in the objective of the study. The authors had described the objective as ‘experiences with perspectives on patient participation in the context of a mental health centre’ – and line 108 shifts it to ‘exploring the experiences of professionals with recovery-orientated practices in mental healthcare’. Is this article providing a portion of data from a larger dataset? If so, make this clear. Could the authors explain the discrepancies in the objective compared with the intention of the focus group questions?

Line 116-7 – do the authors mean content analysis?

Please ensure the terms are clear and consistent with the way the authors describe the participants of the study. Are they healthcare professionals or participants or informants? This clarification will help the reader to understand who is being referred to when communicating the message.

Can the authors explain the connection between experiences with perspectives on patient participation and patient participation is not a free choice?

Line 149 – ‘participation is not an individual choice’ – how has this statement originated from the focus groups? What is the connection between the statement and what the participants discussed?

Lines 174-175 – Can the authors explain more about the interpretation and relate the interpretation to what is evident in the participant's words? It is unclear what the function of the rhetorical question is on line 175, please consider rephrasing this.

Lines 181-182 – this section appears incomplete. Please relate the interpretations to the sub-theme or theme.

Lines 186-188 – could the authors clarify the relationship between the structural methods and the experiences of the patient participation? As a reader I must assume what the relationship is and how it might be relevant to overall themes; misconception can be clarified with explicit identification of the relationship.  

Lines 209-211 – how does this interpretation relate to the theme?

Lines 244 to 252 – please ensure the paragraphs are correctly aligned.

Line 253 - Please provide a more detailed theme description

Line 278-279 – who are the healthcare professionals you are referring to. Who else might value this knowledge?

Line 279 – please clarify if you are referring to healthcare professionals generally or the participants involved in the focus groups.

Line 286 what previous research? What did previous research show and how was it similar and different from the findings in this study?

Lines 287-289 – please explain what is the relevance of the consequence and does the relevance relate to the challenges of incorporating patient participation in a mental healthcare centre.

Lines 298-299 What is the other research are you referring to? Please be more explicit about the research, what they did and found, and also how it relates to your findings.

Line 302 – which study are you referring to?

Lines 314-318 – This interpretation is good, it follows the data and the description of findings.

Line 319 – which approach are the authors referring to when they say ‘this approach’?

Lines 319-324 – what might the problem be with the approach? What interpretation could the authors offer that draws on the data within the study?

Line 325 – as the authors did not include users in your sample, the authors cannot make this explicit claim. Please amend to be accurate. 

Author Response

Dear reviewer

We are very grateful for your thorough work and constructive concrete comments. We agree with your criticisms and have chosen to follow them all in our correction.

I feel as though there is a missed opportunity to comment or contribute to the discussion about healthcare professionals placing the burden on patients who have not been enabled (or have the skills) to make decisions about healthcare. While there the study touches on this idea, there is an opportunity to contribute more explicitly.

We have added a section to the discussion on the ethical obligation of health professionals to offer users influence to the extent they want it.

Overall, the text appears to be underdeveloped. There is a lack of explicit clarity and the relevance of some ideas is not present. The introduction attempts to establish the value of this study, but I am left wondering what the relevance is of some ideas.

We have read our intro again. We try to show the sparse research on patient participation and what consequences this has for the influence the user has. We would like to contribute with updated knowledge about how the health professionals who collaborate with users in practice perceive patient participation. a knowledge that can lead to greater clarifications of what it actually means from the health professionals' perspectives.

The results do well present the content of the focus groups and the commentary on some themes needs to be expanded.

We have gone through all our reflections on the empirical evidence and this has led to an expansion of our reflections. We hope it meets the criticism.

The discussion does well to pull the ideas and themes together.

We have gone through the discussion again and tried to clarify ideas and themes.

Please proofread and ensure all spelling and typos are corrected.

We get a native Englishman to profread it

Line 12 ‘centre’ should be ‘centres’. Corrected

The objective sentence in the abstract does not make sense, the ‘experiences with perspectives on participation’. Please amend this sentence. Corrected

Introduction – Are there specific Western countries that are adopting patient participation in the way that is conceived of in this study? As a Southern Hemisphere healthcare system user (of a Western country), patient participation is not conceptualised in the same way as described in the study.

We cannot be absolutely sure, so we insert a caveat in the sentence

Line 35 – ‘any’ does not make sense in the sentence. Corrected

Line 35-36 – Please clarify the sentences. What do the authors mean by different perspectives and what is the relevance of views that are linked to different discourses? What is the relevance of discourse in relation to understanding patient participation or the study? Could the authors consider the value of the sentences and how this contributes meaning to the paragraph and the wider study?  Corrected

Line 41 ‘users’. Please use terms consistently, are the people being referred to ‘users’ or ‘patients’. Consistency in terms will enable the reader to follow the writing rather than getting caught up on who is the subject. We are agree and will use patient

Line 49 – why is it a vague concept? Please consider clarifying the intended communication in this sentence. We have added an explanation and hope that clears it up

Line 55 – too many spaces. Why would readers, patients, or healthcare professionals care that the participation discourse includes the burden of increased diseases on the state (i.e., public finances and maintaining the welfare state)? Could the authors link the sentences on lines 55 and 56 to the topic of participant discourse? Sentence on line 57 to earlier paragraph on line 56. We have deleted the part of the text you point out

Line 58 too many spaces. Corrected

Line 60 ‘most possible treatment’ does not make sense. Please correct or explain. Corrected

Line 64 – which healthcare policies are the authors referring to? What might these policies have to do with patient participation? We have deleted and moved around so that more sources are added and there is an explanation of what is meant afterwards

Line 67 – what is the meaning ‘and so on’ in this sentence? so on is now deleted

Line 69-67 – the sentence does make sense. What situation basis are the authors referring and which unique situations do they mean? Could they clarify this sentence to communicate the intended meaning more clearly? Please also clarify the subsequent sentence. We have added an explanation to clarify our messages

Line 72 – What research are the authors referring to when they say ‘the research lacks clarity’? Corrected and add more references 

Line 77 – as per an earlier comment, the objective statement of ‘experiences with perspectives on patient participation’ does not make sense. It is unusual to see the combination of ‘experiences with perspectives’ as these are two distinct concepts and perhaps selecting either experiences or perspectives might start to communicate the objective more clearly. Corrected

Line 85 – ‘teacher’ might be a more accessible word than ‘pedagogues’ for western English speakers. Corrected

Line 86 – do the authors mean ‘users’ rather than ‘the user’ and is this ‘the users of the mental health centre’? Corrected

Line 87 – what do the authors mean ‘with each section’? Corrected

Lines 83 and 87 are repeating similar information We have reviewed the entire text and corrected it so that there are no repetitions

Please be consistent and clear about who the actor is in the sampling section. The authors shift from ‘the first author’ to ‘we’ to ‘the first author’ – this is confusing. Corrected

Line 90 start the sentence with ‘Twenty-four participants….’Corrected

Line 94 too many spaces Corrected

Line 98 – the title is misspelt. Focus group should not be plural. Please check the spaces and alignment of the table contents.  Corrected

Line 108 – why is there an apostrophe? Corrected

In lines 108 and 109 there is a shift in the objective of the study. The authors had described the objective as ‘experiences with perspectives on patient participation in the context of a mental health centre’ – and line 108 shifts it to ‘exploring the experiences of professionals with recovery-orientated practices in mental healthcare’. Is this article providing a portion of data from a larger dataset? If so, make this clear. Could the authors explain the discrepancies in the objective compared with the intention of the focus group questions? We could see the change and have fixed it now

Line 116-7 – do the authors mean content analysis? Corrected

Please ensure the terms are clear and consistent with the way the authors describe the participants of the study. Are they healthcare professionals or participants or informants? This clarification will help the reader to understand who is being referred to when communicating the message. Corrected

Can the authors explain the connection between experiences with perspectives on patient participation and patient participation is not a free choice? We have delete perspectives and hope it is more clear now 

Line 149 – ‘participation is not an individual choice’ – how has this statement originated from the focus groups? What is the connection between the statement and what the participants discussed? We have added an explanation below the theme

Lines 174-175 – Can the authors explain more about the interpretation and relate the interpretation to what is evident in the participant's words? It is unclear what the function of the rhetorical question is on line 175, please consider rephrasing this. Agreee and is corrected 

Lines 181-182 – this section appears incomplete. Please relate the interpretations to the sub-theme or theme. We have added more explanation

Lines 186-188 – could the authors clarify the relationship between the structural methods and the experiences of the patient participation? As a reader I must assume what the relationship is and how it might be relevant to overall themes; misconception can be clarified with explicit identification of the relationship.  We have added more explanation

Lines 209-211 – how does this interpretation relate to the theme? Corrected

Lines 244 to 252 – please ensure the paragraphs are correctly aligned. Corrected

Line 253 - Please provide a more detailed theme description Corrected

Line 278-279 – who are the healthcare professionals you are referring to. Who else might value this knowledge? Corrected

Line 279 – please clarify if you are referring to healthcare professionals generally or the participants involved in the focus groups. Corrected 

Line 286 what previous research? What did previous research show and how was it similar and different from the findings in this study? We have add more about this 

Lines 287-289 – please explain what is the relevance of the consequence and does the relevance relate to the challenges of incorporating patient participation in a mental healthcare centre.We have add more about this 

Lines 298-299 What is the other research are you referring to? Please be more explicit about the research, what they did and found, and also how it relates to your findings.We have add more about this 

Line 302 – which study are you referring to? we have add ref

Lines 314-318 – This interpretation is good, it follows the data and the description of findings. thank you 

Line 319 – which approach are the authors referring to when they say ‘this approach’? we have now explain it 

Lines 319-324 – what might the problem be with the approach? What interpretation could the authors offer that draws on the data within the study? we have explain it now

Line 325 – as the authors did not include users in your sample, the authors cannot make this explicit claim. Please amend to be accurate.  corrected 

Round 2

Reviewer 1 Report

My comments were adressed. In my opinion the manuscripts could be published after a light revision of the English style. 

Author Response

Dear Reviewer

We are grateful for your efforts and we will have the article proofread.

Best regards 

all the authors 

Reviewer 2 Report

Thank you for your submitting your revisions quickly. The revisions have led to significant improvements in the clarity and readability of your manuscript. 
There are still a few minor errors remaining (listed below), but I would also recommend thoroughly proofreading the manuscript or using text-to-talk software so you can listen to the delivery of the sentences. Please do not rely on the following list as an absolutely complete list of errors. Grammatical, spelling, and formatting errors may still exist and it is important to resolve all errors to ensure that your paper delivers your intended message. 

- Line 37 - Do not start a sentence with 'E.g.,' - either add this to the sentence prior or begin the sentence with 'For example, the meaning of patient participation...'
- Lines 48, 60, 61 - rather than 'his/her' use the non-binary pronoun of 'their' (...the patient must have information about their illness and treatment').
- Line 57 - opt for gender pronoun 'themself' over '..lies with the person himself'.
- Lines 54 to 69 - uses multiple very short paragraphs please consider joining these paragraphs together or creating a more cohesive passage. 
-Lines 174 to 180 - These lines appear to represent a direct quote from a healthcare professsional, however, the format of the paragraph is the same as the author's commentary. Are you able to indent lines 174 to 180 to indicate a passage of text that is different from the author's writing? (Similar to the format in lines 308 to 310 or 313 to 315. Please use an indented paragraph format consistently for all healthcare professional quotes, as this will help the reader to understand what is quoted (serving as data) and what is commentary by the author. 
Lines 193-195 - This sentence does not make sense. I cannot determine what the intended message of this sentence is, please correct it. 
Line 205 - change himself to a gender neutral term (themself)
Line 218 - insert 'health' - 'Mental health disorders can be experienced...'
Lines 260 to 262 - This sentence does make sense: 'Structural methods mean, e.g., the health professionals' choice of treatment as psychoeducation, cognitive behaviour therapy, or environmental therapy.' - Please finish the sentence 'Structure methods mean...' 
Line 295 - The sentence lacks clarity and it might make more sense to say 'The patient may not be aware of their illness or their need for treatment.'
Line 325 - either remove 'such as' or 'e.g.,'
Line 348 - remove 'i.e.,'
Line 349 - remove 'e.g.,'
Line 350 - which study are you referring to when you say 'As the study shows,'
Lines 341 to 389 - Please break this passage into multiple paragraphs. A Paragraph this long is very hard to read and your intended message is lost. 
Line 437 - opt for a gender neutral pronoun - 'their treatment'
Lines 445-446 and 454-455 repeat the same acknowledgements.

Author Response

Dear reviewer 

We are grateful for your efforts and we have also carefully reviewed all the comments this time and have tried to accommodate them all. We also have proofreading done.

Best regards 

all the authors